# DeFake: Data-Efficient Adaptation for Generalized Deepfake Detection

## Abstract

While deepfake detection methods have seen significant progress, current approaches focus on detecting fully synthetic or partially manipulated images separately, and often rely on large amounts of labeled training data. However, in real world, deepfakes can originate from any paradigm. In this work, we propose a generalized deepfake detection method, *DeFake* (Data-Efficient Adaptation for Generalized Deepfake Detection) which can detect both fully synthetic and partially manipulated images simultaneously. We reframe the generalization problem as a data-efficient adaptation of a base synthetic image detector to the task of partial manipulation detection using limited training samples, without degrading the original synthetic image detection task. We introduce three novel modules: (a) Noise-aware Patch Enhancement (NPE) which captures local manipulation artifacts present in partially manipulated images, (b) Adaptive Score Aggregation (ASA) which modulates the influence of the global image-level semantics and the local patch-level artifacts, and (c) Multi-scale alignment which enhances discriminative learning at both image and patch-level. The proposed modules are generalizable and can be integrated into various base models. Extensive experiments on 14 datasets across both paradigms demonstrate the effectiveness of our proposed DeFake, outperforming state-of-the-art approaches in both settings.

## 1 Introduction

Recent advances in deep learning have enabled generative models, such as GANs and diffusion models, to create photorealistic synthetic images (Masood et al., 2023). While useful for entertainment and digital art, these methods can also be exploited for misinformation and privacy breaches, posing serious threats to media integrity and public trust.

Existing deepfake detectors typically fall into two distinct categories: (1) fully synthetic image detectors, which identify completely generated images (Ojha et al., 2023; Liu et al., 2024), and (2) partially manipulated image detectors, which can detect forgeries in otherwise pristine images (Guillaro et al., 2023; Li et al., 2024). These methods are trained on large-scale datasets and aim to improve generalization to unseen generation models within their respective categories. However, real-world deepfakes can originate from either paradigm, and current detectors overlook generalization across both. As such images are often visually indistinguishable from authentic ones, there is a need for generalized models that can seamlessly detect both fully synthetic and partially manipulated content.

Recently, large-scale pretrained models like CLIP have shown strong potential for fully synthetic image detection. For example, UniFD (Ojha et al., 2023) leveraged CLIP's image encoder to distinguish real from fake images, while later works introduced adapter modules for forgery-specific features (Liu et al., 2024; Chen et al., 2025). For partially manipulated image detection, methods often train decoders on aggregated features to produce pixel-wise segmentation maps (Guillaro et al., 2023; Smeu et al., 2025). These approaches focus on local edge-related artifacts, whereas fully synthetic detection relies on global representations—limiting cross-task generalization (Fig. 2).

To understand this limitation better, we analyze features from three CLIP-based synthetic detectors on a dataset mixing real, fully synthetic (ProGAN), and partially manipulated images (Fig. 1). We observe that synthetic images form a well-separated cluster from real ones, whereas partially manipulated images heavily overlap with real images. Since the majority of patches in such images

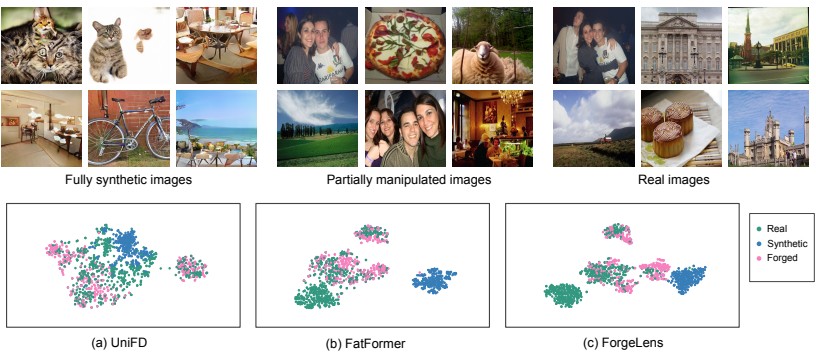

Figure 1: t-SNE visualization of CLIP ViT-L/14 based fully synthetic detectors (a) UniFD (CVPR'23), (b) FatFormer (CVPR'24), (c) ForgeLens (ICCV'25) on a mix of fully synthetic (Pro-GAN), partially manipulated (CASIA1, DSO-1) and real images. These methods demonstrate good separability between real and fully synthetic images, but show considerable overlap between partially manipulated and real images, indicating poor generalization in fine-grained forgery detection.

remain unaltered, the aggregated global embedding is dominated by real patches, diluting the contribution of manipulated regions. Consequently, detectors that rely primarily on global-level cues fail to produce a discriminative separation between partially manipulated and real images. Training a unified model on both categories could, in principle, capture global and local signals. In practice, this is limited by (i) the cost of large-scale pixel-level annotations, (ii) redundant training, as fully synthetic detectors already learn strong holistic forgery cues, partly relevant to manipulated images. The challenge is not to learn entirely new features, but to calibrate the feature space, amplifying manipulated-patch signals while retaining global discriminability.

To this end, we propose DeFake (*Data-Efficient Adaptation for Generalized Deepfake Detection*), where we reframe the problem as a data-efficient adaptation task. Instead of retraining on large combined datasets—which can overwrite the global discriminability acquired from synthetic images—we adapt the CLIP-based model—originally trained for fully synthetic image detection—to the new task of partial manipulation detection using only a few training samples. By restricting adaptation to a small training set, the model is encouraged to preserve its base performance on synthetic detection, while making minimal but targeted adjustments to partially manipulated images.

The proposed framework comprises three novel modules: (1) *Noise-aware Patch Enhancement module*, which guides the CLIP-based encoder to capture subtle tampering artifacts by enriching

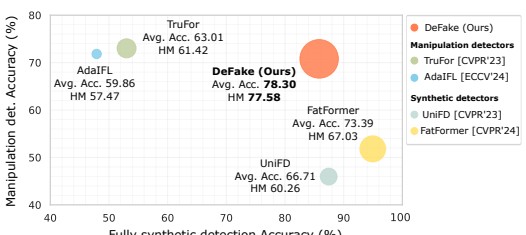

Figure 2: Scatter plot of average accuracy on fully synthetic (x-axis) and partially manipulated (y-axis) datasets. Existing deepfake detectors show limited cross-task generalization, while our proposed approach generalizes across both tasks simultaneously, achieving the highest average accuracy and harmonic mean.

ing visual patch representations through cross-attention fusion with learned noise patterns; (2) *Adaptive Score Aggregation module*, which balances global semantic coherence with local patch-level cues, enabling the model to retain its original capability for synthetic image detection. To achieve this, we employ a lightweight network that dynamically modulates the contributions of global image-level and local patch-level similarity scores based on the input image features. This allows the model to seamlessly handle both fully synthetic images (which require global analysis) and partially manipulated images (which require attention to local artifacts), without catastrophic forgetting; and (3) *Multi-Scale Alignment module*, which uses contrastive learning to enhance discrimination at both the image and patch levels. Our approach significantly outperforms the state-of-the-art for both paradigms, while requiring minimal training data, achieving the highest average accuracy of 78.30% and harmonic mean (HM) of 77.58%. Overall, we make the following contributions:

- We propose a generalized framework, *DeFake*, that can simultaneously detect both fully synthetic and partially manipulated images using minimal training data, by formulating the problem as a data-efficient adaptation task.

- We introduce three novel modules: (i) *Noise-aware Patch Enhancement*, (ii) *Adaptive Score Aggregation* and (iii) *Multi-Scale Alignment* for this task.

- Extensive experiments demonstrate that our approach achieves state-of-the-art performance on both detection tasks, while requiring significantly fewer training examples compared to existing methods.

## 2 RELATED WORK

Deepfake detection research has developed along a few axes: (1) fully synthetic image detection, (2) partial manipulation detection, (3) continual deepfake detection, which we briefly discuss below.

**Fully synthetic image detection:** Early works (Frank et al., 2020; Marra et al., 2018) explored learning-based approaches to detect GAN-generated images using high-frequency methods like DCT. Later works showed that such approaches do not generalize well to other generative models. Zhang et al. (2019) found that different GAN models share common artifacts due to upsampling layers, and leveraged spectra-based classifier. Wang et al. (2020) addressed generalization by training on large-scale data along with augmentations from one GAN model. Tan et al. (2024) introduced CNN layers on FFT magnitude and phase spectrum to learn source agnostic artifacts. Recent works use pretrained VLMs like CLIP (Radford et al., 2021) to leverage the strong pretrained features. UniFD (Ojha et al., 2023) proposed learning a classifier on the frozen CLIP-ViT. FatFormer (Liu et al., 2024) further learns frequency components inside CLIP-ViT along with text guidance to enhance performance. ForgeLens (Chen et al., 2025) extracts forgery features by adding trainable layers in CLIP-ViT blocks and training a transformer on the aggregated layer-wise CLS tokens.

**Partial manipulation detection:** Partially manipulated images can be produced manually (e.g., cheapfakes) or by generation methods. While initial works (Huh et al., 2018; Wen et al., 2016) detected simple copy-move or splicing, later works aimed to generalize to unseen forgeries. Dong et al. (2022) used self-supervised training on $385$ forgeries with an anomaly detector, Liu et al. (2022) used progressive multi-scale feature training, while self-attention mechanism is utilized by Hao et al. (2021) to hierarchically model feature maps at different scales. Recently, AdaIFL (Li et al., 2024) employs a MoE model to dynamically capture forgery traces, and learns a decoder for segmenting the tampered region. Cozzolino & Verdoliva (2019) addresses this by training a Noiseprint model on large-scale camera-image footprints to capture an underlying pattern and any deviation in this pattern signifies a local tampering of the image. Trufor (Guillaro et al., 2023) encodes these Noiseprint features to train a segmentation block, in addition to a classification branch. More recently, Smeu et al. (2025) used CLIP features for training a segmentation network for forgery localization.

**Continual deepfake detection.** With fast-evolving generative models, continual learning methods (Kim et al., 2021; Tian et al., 2024; Laiti et al., 2024) aim to adapt without forgetting. However, they mainly focus on incremental tasks with overlapping domains, and overlook generalization across both synthetic and manipulated images.

*In this work, different from others, we aim to achieve generalization across both synthetic and partially manipulated image classification in a data-efficient manner. Our method allows efficient adaptation to the partial manipulation task using limited training examples, with minimal forgetting of the original synthetic image detection task.*

## 3 MOTIVATION

The deepfake detection problem can be formulated as distinguishing natural images from distribution shifts caused by fully synthetic or partially manipulated content. Fully synthetic and partially manipulated images differ in the extent of manipulation, but share inconsistencies with real image statistics. Formally, let $\mathcal{D}_r, \mathcal{D}_f, \mathcal{D}_p$ denote real, fully synthetic and partially manipulated distributions respectively. A manipulated image $X$ can be modeled as a mixture distribution:

$$P(X|\mathcal{D}_p) = \lambda P(X|\mathcal{D}_f) + (1 - \lambda)P(X|\mathcal{D}_r) \tag{1}$$

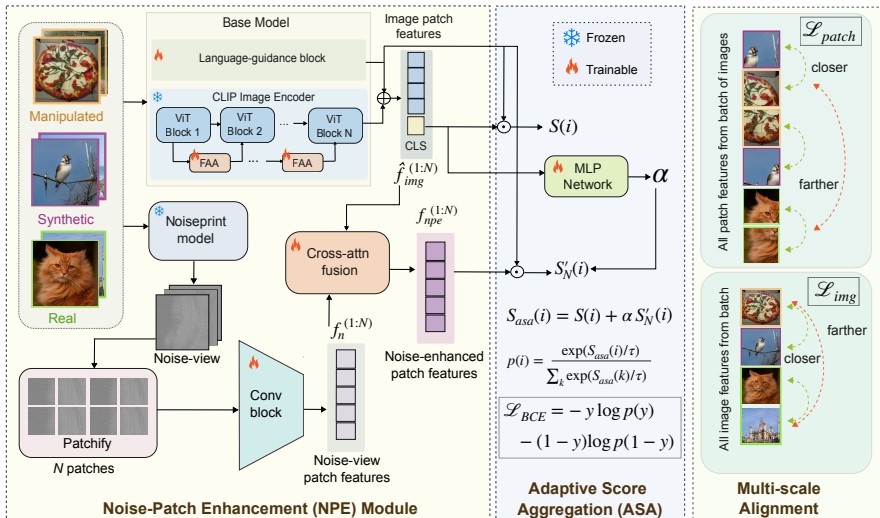

Figure 3: Overview of the proposed DeFake framework. Here, we use the base model as FatFormer which includes the CLIP text and image encoders. Our proposed modules can be integrated to other CLIP-based base models as well. First, in the Noise-Patch Enhancement (NPE) module, we extract noise-views of the input image using a noise-extractor model to obtain noise-view patch features which is used to refine the image patches with nuanced forgery artifacts. The Adaptive Score Aggregation (ASA) module is then used to adaptively combine the image-level and patch-level scores. The framework is trained with Multi-Scale Alignment (MSA), which involves binary cross-entropy loss along with patch-level and image-level contrastive losses.

where, $0 < \lambda < 1$ reflects the proportion of fake patches in the image. The base synthetic detection model is trained to separate between $P(X|\mathcal{D}_f)$ and $P(X|\mathcal{D}_r)$, and hence detecting $P(X|\mathcal{D}_p)$ can be framed as adapting the decision boundary to account for localized mixtures of real and fake patches, rather than training an entirely new classifier.

For partially manipulated images, $\lambda$ is typically small since most patches remain real, causing $P(X|\mathcal{D}_p)$ to overlap with $P(X|\mathcal{D}_r)$. Our idea is to amplify local patch-level inconsistencies with forgery-specific cues, hence increasing the contribution of manipulated regions, shifting $P(X|\mathcal{D}_p)$ closer to $P(X|\mathcal{D}_f)$. However, local cues are sparse or absent in synthetic images, where artifacts are distributed globally. Over-reliance on patch-level embeddings can dilute global signals, reducing the discriminability of $\mathcal{D}_f$ from $\mathcal{D}_r$, leading to degradation of synthetic detection task. To mitigate this, we adaptively reweight global and local similarity scores, so that local manipulations are highlighted without degrading global discriminability.

## 4 PROBLEM FORMULATION

Here, we introduce the problem setup, followed by describing the base framework. Let $\Theta$ denote a pre-trained base framework trained for synthetic image detection. Given a relatively small dataset of locally manipulated images $\mathcal{D} = \{(X_i, y_i, \mathcal{M}_i)\}_{i=1}^{K}$, our objective is to adapt $\Theta$ using $\mathcal{D}$, while retaining its original performance on fully synthesized images. Here, $X_i, y_i$ and $\mathcal{M}_i$ denote the image, label $\{1 \text{ (fake)}, 0 \text{ (real)}\}$ and the corresponding binary ground truth mask respectively.

**Base framework.** Inspired by the success of CLIP-based synthetic image detectors, we adopt the recently proposed FatFormer architecture (Liu et al., 2024) as our base model. We also show that our proposed modules generalize to other base models as well. For the task of synthetic image detection, FatFormer incorporates forgery-aware adapter (FAA) modules into the encoder blocks of a frozen CLIP ViT. These adapters aggregate frequency-rich signals by learning grouped attention over the Discrete Wavelet Transforms (DWT) of the internal feature representations. Additionally, it introduces a language-guided alignment (LGA) module which achieves enhanced prompts $\hat{p}_{ctx}$ by

conditioning learnable prompt vectors with the $N$ image patch features. These enhanced prompts are appended to the classnames (here, *real* and *fake*) and fed to the text encoder to obtain text features. Subsequently, it also obtains aligned image patch features $\hat{f}_{img}^{(1:N)}$ by learning a cross-attention module between text features and the image patches. Finally, the classification is performed using CLIP similarity score $S(i) = cos(f_{img}^{[CLS]}, f_{text}^i)$ and the mean of aligned patch similarity scores $S'(i) = \frac{1}{N} \sum_{j=1}^{N} cos(\hat{f}_{img}^j, f_{text}^i)$, where $f_{text}^i$ denotes the $i^{th}$ text feature and $i \in \{0, 1\}$.

## 5 THE PROPOSED DEFAKE FRAMEWORK

We introduce the proposed framework, DeFake (Fig. 3), which enables adaptation to the new task of partial manipulation detection using only limited training samples, while retaining performance on the base task of fully synthetic image detection. DeFake comprises three key components: 1) Noise-aware Patch Enhancement (NPE): This module enriches image patches by incorporating learnable noise-view patches, designed to capture local tampering artifacts crucial for the forgery detection task, 2) Adaptive Score Aggregation (ASA): A lightweight image-conditioning network modulates the influence of global image-level semantics and local patch-level signals. This helps preserve performance on the base task by mitigating catastrophic forgetting. 3) Multi-Scale Alignment (MSA): This component introduces complementary alignment objectives at both the image and patch levels, enabling consistent learning across global and local representation spaces.

### 5.1 NOISE-AWARE PATCH ENHANCEMENT MODULE

Partially manipulated images often contain low-level inconsistencies and blending artifacts that do not significantly affect the global semantics of the image. As a result, recent CLIP-based detectors—designed primarily for fully synthetic image detection and relying on CLIP as a feature extractor for classification—struggle to identify such localized manipulations, as shown in Fig. 1. Several works (Cozzolino & Verdoliva, 2019; Guillaro et al., 2023; Zhang et al., 2025) have demonstrated the effectiveness of using additional noise-based views of the input image, rather than relying solely on the RGB domain, for detecting local manipulations. These noise views can expose subtle deviations introduced during tampering, which may remain imperceptible in the RGB space. However, these works typically train dedicated encoder-decoder networks on RGB and noise inputs to predict pixel-level masks, requiring large-scale annotated datasets. In contrast, our goal is to use noise views to refine CLIP's patch representations, avoiding full encoder training while enabling adaptation with only a few manipulated samples.

Given an input image $X \in \mathbb{R}^{3 \times H \times W}$, we extract the noise-view map $X_n \in \mathbb{R}^{H \times W}$, using the Noiseprint++ noise-extractor model (Guillaro et al., 2023), where $H$ and $W$ denote the height and width of the image, respectively. This noise-view map is then partitioned into $N = HW/P^2$ non-overlapping patches each of size $P \times P$, denoted as $X_n^{(1:N)}$. These noise-view patches are subsequently passed through a learnable block to obtain the corresponding noise-view patch features as follows:

$$f_n^{(1:N)} = \text{Conv}\left(X_n^{(1:N)}\right) \tag{2}$$

where, $f_n^{(1:N)} \in \mathbb{R}^{N \times d}$, Conv denotes a set of convolutional blocks followed by ReLU, and $d$ denotes the embedding dimension of CLIP. Next, given the output image patch features $\hat{f}_{img}^{(1:N)} \in \mathbb{R}^{N \times d}$ from the CLIP encoder, we introduce a cross-attention fusion module that enriches each image patch by attending to its corresponding noise-view patch features, formulated as:

$$Q = W_Q \hat{f}_{img}^{(1:N)}, \ K = W_K f_n^{(1:N)}, \ V = W_V f_n^{(1:N)}$$
$$f_{npe}^{(1:N)} = LN\left(\text{softmax}\left(QK^T/\sqrt{d}\right)V\right) \tag{3}$$

where, $W_Q, W_K, W_V \in \mathbb{R}^{d \times d}$ are learnable projections and LN denotes Layer Normalization. The output $f_{npe}^{(1:N)} \in \mathbb{R}^{N \times d}$ represents the noise-enhanced patch features.

### 5.2 ADAPTIVE SCORE AGGREGATION

After enhancing the image patch features to aid the forgery detection task, we now turn our attention to preserving the base model's performance on synthetic image classification. The classification

of fully generated images typically relies on both the global image feature corresponding to the CLS token and the aggregated patch-level features. Liu et al. (2024) demonstrated improved performance by adding the CLIP similarity score $S(i)$ and the mean patch similarity scores $S'(i)$ as described earlier. Therefore, it is essential to ensure that the patch-level modifications introduced in the previous module do not degrade the performance of the base task. Following the refinement of patch features using the noise-view information, the mean similarity score can now be reformulated as: $S'_N(i) = \frac{1}{N} \sum_{j=1}^{N} cos(f_{npe}^j, f_{text}^i)$, for $N$ patches, where, $f_{text}^i$ denotes the $i^{th}$ text feature, and $f_{npe}$ denotes the noise-enhanced patch features from eqn. (3). To better balance the trade-off between detecting fully synthetic and partially manipulated images, we propose dynamically modulating the contributions of the global and patch-level similarity scores based on the input image itself. Specifically, we project the input image feature to obtain a scaling weight as follows:

$$\alpha(z) = \sigma\left(W_1\left(\text{ReLU}\left(W_2 . z\right)\right)\right) \tag{4}$$

where, $z = f_{img}^{[CLS]} \in \mathbb{R}^d$ is the $d$-dimensional input image feature, $W_2 \in \mathbb{R}^{d \times d'}, W_1 \in \mathbb{R}^{d' \times 1}$ denote projection layers and $\sigma$ denotes the Sigmoid function. The similarity scores are then adaptively aggregated as:

$$S_{asa}(i) = S(i) + \alpha\, S'_N(i) \tag{5}$$

for $i \in \{0, 1\}$. This enables the model to adapt the decision based on the given input.

### 5.3 MULTI-SCALE ALIGNMENT

Since training samples are limited, we utilize all the patches from each image during training, allowing the model to benefit from dense localized supervision. This not only enhances learning through additional patch-level guidance but also encourages the model to rely less on object-specific semantics and more on intrinsic manipulation cues. Specifically, we introduce a patch-wise contrastive loss that encourages alignment among patches of the same class (pristine or manipulated) while enforcing separation between them. We divide the ground truth binary mask $\mathcal{M}$ into $N$ non-overlapping patches and label them as fake if at least one fake pixel is present in them. Patch-wise loss is computed collectively for all image patches within a batch. Additionally, to prevent bias towards the real class, we perform a balanced sampling of real and fake patches in each training batch, since the real patches can often exceed fake patches for localized manipulations. The resulting patch-wise loss for the $i^{th}$ patch can be formulated as follows:

$$\mathcal{L}_{patch} = -\log\left(\frac{\exp(\hat{f}_{img}^{(i)} \cdot \hat{f}_{img}^{(j)}/\tau)}{\sum\limits_{\substack{k=1 \\ k \neq i}}^{|NB|} \exp(\hat{f}_{img}^{(i)} \cdot \hat{f}_{img}^{(k)}/\tau)}\right) \tag{6}$$

where, $\hat{f}_{img}^{(i)}$ and $\hat{f}_{img}^{(j)}$ refer to the image patch representations from the same class (real/fake), $\tau$ denotes the temperature constant, and $B$ is the batch size.

Additionally, in order to learn a robust representation space with few training samples, we introduce an image-wise contrastive loss which aligns embeddings of real images and fake images, while separating them from each other. The loss for a given training sample is given as:

$$\mathcal{L}_{img} = -\log\left(\frac{\exp(f_{img}^{i\,[CLS]} \cdot f_{img}^{j\,[CLS]}/\tau)}{\sum\limits_{\substack{k=1 \\ k \neq i}}^{|B|} \exp(f_{img}^{i\,[CLS]} \cdot f_{img}^{k\,[CLS]}/\tau)}\right) \tag{7}$$

where, $f_{img}^{i\,[CLS]}, f_{img}^{j\,[CLS]}$ belong to the same class, and $B$ denotes the batch size. The introduction of both these losses enables a hierarchical alignment framework which jointly structures patch-level and image-level representation spaces.

**Overall Objective:** The final objective for training the base parameters and the proposed modules consists of the binary cross-entropy (BCE) loss applied on the adaptively aggregated score $S_{asa}$ along with the image-wise and patch-wise loss functions. Given, prediction $\hat{p}(i) = \exp(S_{asa}(i)/\tau)/\sum_k \exp(S_{asa}(k)/\tau)$, the BCE loss is $\mathcal{L}_{BCE} = -y \log \hat{p}(y) - (1-y) \log \hat{p}(1-y)$. The final objective can be written as:

$$\mathcal{L} = \lambda_1 \mathcal{L}_{BCE} + \lambda_2 \mathcal{L}_{patch} + \lambda_3 \mathcal{L}_{img} \tag{8}$$

where, $\lambda_1, \lambda_2, \lambda_3$ weighs the different loss counterparts.

| Method | Fully Synthetic Image Datasets | | | | | | | | Partially Manipulated Image Datasets | | | | | | | | HM ↑ | Avg (T) ↑ |
|---|---|---|---|---|---|---|---|---|---|---|---|---|---|---|---|---|---|---|
| | ProGAN | BigGAN | GauGAN | StarGAN | Deepfake | DALLE | Guided | Avg (F) | CASIA1 | Columbia | Coverage | NIST16 | DSO-1 | CocoGLIDE | MagicBrush | Avg (P) | | |
| *Synthetic Det.* | | | | | | | | | | | | | | | | | | |
| UniFD (CVPR'23) | 99.75 | 93.40 | 99.09 | 96.77 | 73.80 | 80.55 | 68.80 | 87.45 | 55.03 | 47.14 | 50.00 | 20.33 | 51.00 | 61.04 | 37.22 | 45.97 | 60.26 | 66.71 |
| FreqNet (AAAI'24) | 99.60 | 91.20 | 92.94 | 84.22 | 92.12 | 97.65 | 71.35 | 89.87 | 48.83 | 48.57 | 49.50 | 42.32 | 51.50 | 50.00 | 35.58 | 46.61 | 61.39 | 68.24 |
| ForgeLens (ICCV'25) | 99.83 | 97.60 | 98.47 | 91.82 | 90.93 | 94.45 | 70.55 | 91.95 | 61.50 | 73.53 | 51.56 | 19.97 | 57.81 | 70.12 | 69.13 | 57.66 | 70.88 | 74.81 |
| FatFormer (CVPR'24) | 99.89 | 98.98 | 98.61 | 99.47 | 92.04 | 98.45 | 77.55 | 95.00 | 47.55 | 66.79 | 50.00 | 26.51 | 53.00 | 70.80 | 47.86 | 51.79 | 67.03 | 73.39 |
| *Partial Det.* | | | | | | | | | | | | | | | | | | |
| Trufor (CVPR'23) | 50.88 | 53.58 | 50.18 | 69.26 | 55.12 | 47.10 | 44.90 | 53.00 | 81.04 | 97.50 | 69.00 | 25.76 | 94.50 | 64.36 | 78.90 | 73.01 | 61.42 | 63.01 |
| AdaIFL (ECCV'24) | 50.54 | 48.15 | 49.71 | 27.89 | 54.19 | 53.55 | 51.25 | 47.90 | 88.66 | 85.71 | 78.50 | 61.75 | 73.13 | 63.57 | 51.45 | 71.82 | 57.47 | 59.86 |
| *Finetuned / Both* | | | | | | | | | | | | | | | | | | |
| HiFi-Net (CVPR'23) | 86.85 | 89.75 | 69.00 | 98.02 | 58.82 | 84.05 | 72.75 | 79.89 | 46.40 | 43.21 | 59.00 | 34.04 | 50.50 | 59.77 | 61.02 | 50.56 | 61.93 | 65.23 |
| Trufor-FT | 49.70 | 53.30 | 50.64 | 53.75 | 46.96 | 48.40 | 48.30 | 50.15 | 51.46 | 52.14 | 51.50 | 37.65 | 47.00 | 57.23 | 39.11 | 48.01 | 49.06 | 49.08 |
| AdaIFL-FT | 49.33 | 49.18 | 48.96 | 49.60 | 50.14 | 46.80 | 44.95 | 48.42 | 63.08 | 86.43 | 59.00 | 53.46 | 69.15 | 58.89 | 47.23 | 62.46 | 54.55 | 55.44 |
| UniFD-FT | 99.18 | 94.75 | 98.45 | 93.20 | 78.65 | 85.80 | 74.20 | 89.18 | 60.41 | 56.76 | 51.00 | 24.10 | 52.00 | 66.41 | 40.37 | 50.01 | 64.08 | 69.59 |
| FreqNet-FT | 99.60 | 91.17 | 92.94 | 84.27 | 92.16 | 97.65 | 71.35 | 89.88 | 50.18 | 56.79 | 51.00 | 42.62 | 52.00 | 51.07 | 42.00 | 49.38 | 63.74 | 69.63 |
| ForgeLens-FT | 99.74 | 88.78 | 95.99 | 69.26 | 72.03 | 88.70 | 67.65 | 83.16 | 69.30 | 88.93 | 53.00 | 27.86 | 62.69 | 71.39 | 67.32 | 62.93 | 71.64 | 73.05 |
| FatFormer-FT | 99.76 | 94.03 | 91.55 | 78.36 | 56.39 | 91.90 | 79.10 | 84.44 | 71.16 | 92.50 | 53.00 | 54.52 | 59.00 | 77.44 | 60.96 | 66.94 | 74.68 | 75.69 |
| **DeFake** (Ours) | 98.96 | 91.72 | 87.54 | 81.54 | 66.05 | 91.60 | 83.30 | 85.82 | 69.88 | 93.21 | 53.00 | 59.94 | 73.50 | 77.34 | 68.64 | 70.79 | **77.58** | **78.30** |

Table 1: Performance comparison of DeFake with state-of-the-art methods across both fully-synthetic and partial manipulation detection datasets. Methods are categorized into Synthetic Det. (synthetic detectors), Partial Det. (partial manipulation detectors) and Finetuned/Both, which includes baseline models adapted on the few training samples or models trained specifically for both tasks (e.g., HiFi-Net). Avg(F) and Avg(P) denote the average accuracy across fully synthetic datasets and manipulated datasets respectively, while Avg(T) denotes the overall accuracy across both tasks. DeFake outperforms all methods in terms of harmonic mean (HM) and Avg (T), shown in bold.

# 6 EVALUATION

## 6.1 EXPERIMENTAL SETUP

**Datasets.** To extensively evaluate the proposed DeFake, we use a variety of standard benchmark datasets as follows.

*i) Synthetic image datasets:* The base fully synthetic image detectors are already pretrained on ProGAN (Karras et al., 2017) generated synthetic images. Here, we consider only a small number of them as replay data during the adaptation. For testing, we evaluate across a diverse set of GAN and diffusion-generated images, including ProGAN, BigGAN (Brock et al., 2018), GauGAN (Park et al., 2019), StarGAN (Choi et al., 2018), Deepfake (Rossler et al., 2019), DALLE (Ramesh et al., 2021), and Guided Diffusion (Dhariwal & Nichol, 2021).

*ii) Partially manipulated datasets:* For adapting the base model to the partial manipulation detection task, we use limited samples from the training datasets introduced in Guillaro et al. (2023), containing both pristine and manipulated images with ground truth masks. For testing, we use pristine images and their manipulated counterparts generated using both cheapfake methods: CASIA1 (Dong et al., 2013), Columbia (Hsu & Chang, 2006), Coverage (Hsu & Chang, 2006), NIST16 (Guan et al., 2019), DSO-1 (De Carvalho et al., 2013) and diffusion models: CocoGLIDE (Guillaro et al., 2023) and MagicBrush (Zhang et al., 2023).

We use a total of 128 training images, comprising 30 replay samples from ProGAN, and the remaining from the partially manipulated datasets. These images are uniformly sampled to ensure an equal number of real and fake samples. We observed no notable gains in performance with slight increase in the training data. Additional dataset details are provided in the appendix.

**Implementation Details and Evaluation Metrics.** We use CLIP ViT-L/14 as the base model, with an input resolution of $224 \times 224$ to remain consistent with CLIP's design. To extract noise-patch features, we employ a 2-layer convolutional block with ReLU activations. It is trained for 11 epochs using AdamW optimizer, with a learning rate of $5 \times 10^{-4}$ for the base modules (FAA and LGA) and $2 \times 10^{-5}$ for the proposed modules. The loss weights are set to $\lambda_1 = 1$, $\lambda_2 = 1$, and $\lambda_3 = 5$, and the temperature constant $\tau = 0.07$. All experiments are conducted using PyTorch on a single NVIDIA RTX A6000 GPU.

We report *Average Accuracy* and *Harmonic Mean (HM)* as our primary evaluation metrics, capturing performance across both the original (synthetic image detection) and new (partial manipulation detection) tasks. Detailed results are given in the appendix.

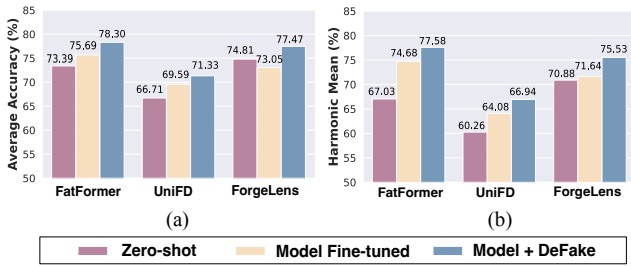

Figure 4: Our proposed modules can be integrated with other CLIP-based base synthetic detectors. DeFake integrated with FatFormer, UniFD and ForgeLens shows improvement over zero-shot as well as standard finetuning performances in terms of (a) avg accuracy, (b) harmonic mean.

## 6.2 COMPARISON WITH STATE-OF-THE-ART

A comprehensive evaluation of the proposed DeFake, along with comparisons to current state-of-the-art methods, is presented in Table 1. To the best of our knowledge, HiFi-Net Guo et al. (2023) is the only prior general-purpose model trained from scratch on large-scale data and evaluated on both fully synthetic and partially manipulated image detection paradigms. In addition, we construct strong baselines by selecting state-of-the-art models tailored to each task, all pre-trained on large-scale datasets.

For *fully synthetic image detection*, we consider: CNN-based (1) Freq-Net (AAAI'24) and CLIP-based (2) UniFD (CVPR'23), (3) Fat-Former (CVPR'24), and (4) Forge-Lens (ICCV'25). For *partially manipulated image detection*, we include two recent methods: (1) Tru-For (CVPR'23), and (2) AdaIFL (ECCV'24). To evaluate adaptability under limited supervision, we fine-tune these baselines using our small

Table 2: Ablation study of the different components of De-Fake shows the importance of each module. Here, Avg(F), Avg(P): average accuracies on fully synthetic and partial manipulation detection tasks.

| NPE | ASA | MSA | | Avg (F) | Avg (P) | HM |
|-----|-----|-----|---|---------|---------|-----|
| – | – | – | (FatFormer-FT) | 84.44 | 66.94 | 74.68 |
| ✓ | – | – | | 83.86 | 67.49 | 74.79 |
| ✓ | ✓ | – | | 85.59 | 68.70 | 76.22 |
| ✓ | ✓ | ✓ | $\mathcal{L}_{BCE} + \mathcal{L}_{patch}$ | 85.58 | 69.70 | 76.83 |
| ✓ | ✓ | ✓ | $\mathcal{L}_{BCE} + \mathcal{L}_{img}$ | **86.80** | 69.80 | 77.38 |
| ✓ | ✓ | ✓ | $\mathcal{L}_{BCE} + \mathcal{L}_{patch} + \mathcal{L}_{img}$ (**DeFake**) | 85.82 | **70.79** | **77.58** |

training set (FT in Table 1). We use the same hyperparameters across all the fine-tuning experiments for fairness. We make the following observations from Table 1:

(i) As expected, fully synthetic image detectors perform well on their intended task but generalize poorly to partially manipulated image detection. Conversely, partial manipulation detectors exhibit the opposite trend, excelling at their own task but struggling with synthetic image detection.

(ii) Fine-tuning (FT) consistently enhances cross-task performance, highlighting the value of adaptation even in low-data scenarios. However, this comes at the cost of reduced performance on the original task. For instance, FatFormer, after being fine-tuned, improves the average zero-shot accuracy on the partially manipulated data from 51.79% to 66.94%, while its original performance on the fully synthetic detection task reduces from 95% to 84.44%.

(iii) Our proposed DeFake significantly outperforms all baselines across both tasks, achieving an average accuracy of 78.30%, which corresponds to a relative improvement of +2.61% over the best-performing method. It also attains a harmonic mean (HM) of 77.58%, a relative gain of +2.90%.

## 6.3 ADDITIONAL ANALYSIS

**1) Ablation Study:** Table 2 presents the ablation study of DeFake. We add our proposed modules—Noise-aware Patch Enhancement (NPE), Adaptive Score Aggregation (ASA), and Multi-Scale Alignment (MSA)—to the base architecture and report harmonic mean (HM), and average accuracies on fully synthetic (Avg(F)) and partial manipulation (Avg(P)) tasks. NPE and ASA improve both tasks by enhancing patch-level forgery detection while strengthening base synthetic performance. Adding the patch-level loss $\mathcal{L}_{patch}$ further boosts Avg(P), whereas the image-level loss $\mathcal{L}_{img}$ improves Avg(F). Integrating all modules and losses yields the best overall performance.

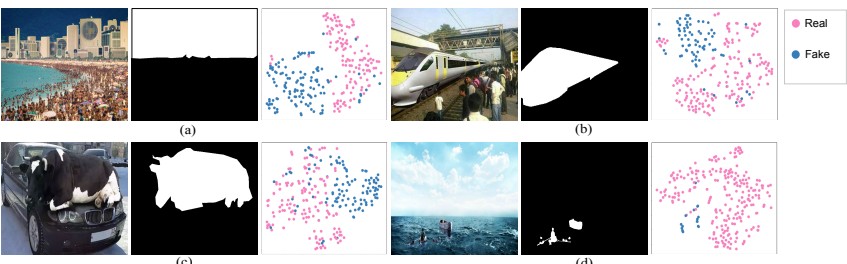

Figure 5: Illustration of some partially manipulated images ((a) - (d)) along with their ground truth masks. Here, we observe the patch-level representation space using t-SNE, corresponding to the fake and real patches in each image. We observe that the fake patches corresponding to the manipulated regions (white in mask) get separated from the real patches.

**2) Generalization to other base models:** While we considered FatFormer as the base model for all the experiments, our proposed modules are generalizable to other fully synthetic image detection models as well. We consider two additional state-of-the-art CLIP-based methods: UniFD and ForgeLens as the base models, and describe the adaptation designs below.

*(a) UniFD* [CVPR 2023]: The base model trains a linear layer on the CLIP image feature for real vs fake classification. First, we extract the noise-view of the input image, and obtain noise patches using the learnable convolutional blocks (eqn.(2)). Next, we incorporate the cross-attention fusion between these noise patch and image patch features from the CLIP encoder, similar to the proposed NPE module to obtain the noise-enhanced patch features. These features are then fed to a lightweight network to get a logit corresponding to the patch representations. Finally, the original logit from the linear layer and the patch-specific logit are adaptively aggregated by conditioning on the input image feature using the proposed ASA module. The whole framework is trained using the multi-scale contrastive learning objectives.

*(b) ForgeLens* [ICCV 2025]: The base framework consists of training bottleneck layers within the CLIP image encoder, followed by training a transformer on the layer-wise CLS tokens from the encoder, to capture multi-stage forgery features effectively. A linear classifier is then trained on the output CLS token from this transformer for the binary classification task. Here also, we obtain the noise-enhanced patch features using the NPE module to enhance the patch-level representations with the fine-grained manipulation artifacts. The proposed ASA module is used to adaptively modulate the original logit (from the base transformer) and the patch-based logit to obtain the final classification logit. Finally, we use the multi-scale alignment objective for training.

As shown in Fig. 4, our method consistently outperforms all base models—both zero-shot and fine-tuned settings—in terms of average accuracy across all datasets and harmonic mean. This highlights the generalizability of our proposed modules in adapting diverse base models for detecting both synthetic and manipulated images simultaneously.

**4) Qualitative Analysis** For further validation of our proposed method, we illustrate the patch-level feature space using t-SNE (Maaten & Hinton, 2008) in Fig. 5. Here, we show some partially manipulated samples, and apply t-SNE on the noise-enhanced patch features to visualize the effect on the patch-level representation space. We observe that the fake patches correspond to distinct clusters compared to real patches, showing the effectiveness of our approach.

**5) Limitations and future scope:** While DeFake is highly effective in low-data regimes, it may not match the performance of models trained from scratch on large-scale datasets if available. Our approach also currently does not perform explicit spatial localization of manipulations. We aim to address these in future work by exploring hybrid classification-localization strategies.

# 7  CONCLUSION

We proposed a generalized framework for deepfake detection that outperforms specialized methods on both synthetic and manipulated images. We reformulated generalization as a data-efficient adaptation problem, showing that a base synthetic detector can be adapted to manipulation detection with minimal data while retaining its original capability. We hope this work inspires future research on practical, generalized, and data-efficient detection frameworks.

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

# A APPENDIX

## A.1 DATASET DETAILS

**Fully synthetic datasets**: The CLIP-based fully synthetic detection models typically use ProGAN-generated synthetic images across 4-classes (car, cat, chair, horse) for training. Here, we take the base model as a fully synthetic detector, and use only a few images from the ProGAN data as replay examples as mentioned in the main paper. For evaluation, we consider images from unseen generation models. From GAN-generated datasets, we take ProGAN, BigGAN (Brock et al., 2018), GauGAN (Park et al., 2019), StarGAN (Choi et al., 2018), Deepfake (Rossler et al., 2019), while for diffusion-generated datasets, we take DALLE (Ramesh et al., 2021) and Guided Diffusion (Dhariwal & Nichol, 2021).

**Partially manipulated datasets**: We use the standard datasets used by partial manipulation detectors (Kwon et al., 2021) for training our model. These include: (a) CASIA-v2, a widely used dataset containing real and forged images with diverse manipulation types; (b) Fantastic Reality, which consists of spliced versions of real images; (c) IMD2020, which reflects real-world data, since it is collected from the internet and social media with practical limitations such as compression artifacts, and (d) Tampered COCO, which is created by copy-move or spicing forgeries on the authentic COCO image dataset. For evaluation, we use CASIA1 (Dong et al., 2013), Columbia (Hsu & Chang, 2006), Coverage (Wen et al., 2016), NIST16 (Guan et al., 2019), DSO-1 (De Carvalho et al., 2013) which are created manually containing copy-move or spicing forgeries. Additionally, we consider two more challenging diffusion-edited datasets, CocoGLIDE (Guillaro et al., 2023) and MagicBrush (Zhang et al., 2023), which contain object-level manipulations using text-based diffusion models. We use the same testing split as followed by Guillaro et al. (2023). For NIST16 we evaluate on the full test set, instead of the smaller subset considered by others. All the datasets contain the authentic (except TampCOCO) and manipulated image counterparts, along with the ground truth binary segmentation maps, where the white pixels correspond to manipulated regions and black denotes the pristine regions.

Contrary to the existing partially manipulated datasets which are trained from scratch on these datasets, we use only a few samples from these datasets for adapting our base model. Specifically, we randomly sample 128 training samples, equally distributed into 64 real and 64 fake images. The complete details of the training data used from each of these datasets are provided in Table 5. We observe that, while reducing the number of training examples results in a drop in average performance, a marginal increase in the training data does not necessarily provide any notable improvement in overall accuracy (Table 4).

## A.2 BASELINES AND FINETUNING DETAILS

In the main paper, we denote methods with suffix "-FT" to indicate that we performed additional finetuning of these pretrained models on the small training set consisting of both the ProGAN replay samples as well as the partially manipulated datasets. Here, we elaborate the details of finetuning protocols for each of these methods.

*(i) UniFD-FT*: Similar to the original work by Ojha et al. (2023), we finetune the final linear classification layer on top of the CLIP-ViT using this dataset, keeping the encoder frozen.

*(ii) FatFormer-FT*: We keep the CLIP image and text encoders frozen, and finetune the forgery-aware adapter (FAA) and language-guided alignment (LGA) blocks similar to the original method (Liu et al., 2024).

*(iii) ForgeLens-FT*: The original model (Chen et al., 2025) consists of a two-stage training process: first, Weight-Shared Guidance Modules (WGSM) are trained inside the CLIP-ViT encoder to extract forgery-related artifacts; next, a transformer model is trained on the concatenated layer-wise CLS tokens from the ViT-blocks, along with a learnable *focus-CLS* token which captures multi-stage attention cues. This *focus-CLS* token is finally used for binary classification by training a linear classifier on top of it. In our finetuning setup, we update the *focus-CLS* token, the transformer model and the final classifier layer, while keeping the base CLIP model frozen, for the final classification task.

*(iv) FreqNet-FT*: The original architecture is based on CNNs (Tan et al., 2024), consisting of frequency convolutional blocks which learns high-frequency contents in the image from both amplitude and phase-spectrum, followed by training a classifier on the features for binary classification. In our setting, we finetune this classifier for the classification task, to avoid overfitting on the small training set.

*(vi) AdaIFL-FT*: The original model employs a mixture-of-experts design to capture diverse manipulation patterns, with importance-aware feature aggregation integrated within the encoder blocks. A dedicated decoder is trained on the aggregated features to localize manipulated regions (Li et al., 2024). We fine-tune the entire decoder module for adaptation, which leverages the forgery-relevant features extracted by the pretrained encoder. To convert the pixel-wise localization output to an image-level classification label, we compute the overlap between the predicted segmentation mask and the ground truth binary mask. If at least $15\%$ of the predicted pixels align with the ground truth manipulated region, we treat it as a correct classification.

*(v) TruFor-FT*: TruFor (Guillaro et al., 2023) addresses both localization and classification of partially manipulated images. Its architecture comprises a Segformer encoder, followed by two decoders: an anomaly decoder that predicts a manipulation segmentation map, and a confidence map decoder that outputs pixel-wise confidence scores. For the classification part, a forgery-detector module is trained, which takes in these two outputs, and performs an image-level binary classification (real or fake). For our finetuning setup, we update the weights of the forgery-detector module only using the small training set for classification, keeping the segmentation branch frozen.

## A.3 DETAILED PERFORMANCE RESULTS

**Main results**: Here, we provide a more comprehensive comparison with state-of-the-art methods across both tasks in Table 6. Along with the average accuracies of the base synthetic detection task Avg(F) and the partial manipulation detection task Avg(P), we also report the real and fake accuracies in this table. Real accuracy indicates the model's ability to correctly classify authentic images as real, while fake accuracy demonstrates the ability to identify partially manipulated or synthetic images effectively. We observe that in terms of classifying fake images, we outperform both the base fully synthetic detection methods as well as the partial manipulation detectors, while using very few training samples for adaptation.

**Generalization to other base models**: As mentioned, our proposed modules are generalizable to alternate CLIP-based synthetic detection models as well, for example, the recently proposed UniFD and ForgeLens. The detailed performance evaluation including average fully synthetic detection accuracy Avg(F), partial manipulation detection accuracy Avg(P), along with the overall

accuracy Avg(T) and harmonic mean (HM) is given in Table 7. We observe that incorporating our proposed modules results in improved performance over standard fine-tuning and zero-shot settings.

### A.4   HYPERPARAMETER SENSITIVITY

We show the effect of changing the weights for the different loss components in the Multi-scale Alignment module in Table 3. Here, $\lambda_1, \lambda_2$ and $\lambda_3$ corresponds to the binary cross-entropy (BCE) loss $\mathcal{L}_{BCE}$, the patch-level loss $\mathcal{L}_{patch}$, and the global image-level loss $\mathcal{L}_{img}$ respectively. While we observe no significant deviations in performance (avg acc and HM) by varying these weights, marginal improvements can be seen when a higher weightage is given to $\mathcal{L}_{img}$.

**Use of Large Language Models (LLMs):** Certain parts of the manuscript, including sentence polishing, paraphrasing, and clarity improvements, were assisted using ChatGPT by OpenAI. All scientific content, ideas, and results remain the authors' own.

Table 3: The effect of varying the weights of the different loss counterparts in the Multi-scale Alignment module.

| $\lambda_1$ | $\lambda_2$ | $\lambda_3$ | Avg (T) (%) | HM (%) |
|---|---|---|---|---|
| 1 | 1 | 2 | 78.03 | 77.17 |
| 5 | 1 | 5 | 77.23 | 76.36 |
| 1 | 5 | 1 | 76.33 | 75.42 |
| 1 | 1 | 5 | 78.30 | 77.58 |

Table 4: Effect of varying training data.

| # Train samples | Avg(T) | HM |
|---|---|---|
| 100 | 74.94 | 74.20 |
| 128 | 78.30 | 77.58 |
| 156 | 78.31 | 77.39 |

Table 5: Number of training images randomly sampled from each dataset. We use much smaller training data compared to existing partial manipulation detection methods.

| Datasets | Ours | TruFor, AdaIFL |
|---|---|---|
| CASIA2 | 25/12 | 7K/5K |
| Fantastic Reality | 14/13 | 16K/19K |
| IMD20 | 9/12 | 414/2K |
| TampCOCO | 0/13 | 0/400 |
| TampRAISE | - | 24K/400K |
| ProGAN | 16/14 | - |
| Real/Fake (Total) | 64/64 | 49K/82K |

| Methods | Acc (%) | **Fully Synthetic image datasets** | | | | | | | | **Partially Manipulated image datasets** | | | | | | | | HM↑ | Avg (T)↑ |
|---|---|---|---|---|---|---|---|---|---|---|---|---|---|---|---|---|---|---|---|
| | | ProGAN | BigGAN | GauGAN | StarGAN | Deepfake | DALLE | Guided | Avg (F) | CASIA1 | Columbia | Coverage | NIST16 | DSO-1 | CocoGLIDE | MagicBrush | Avg (P) | | |
| UniFD | real | 99.58 | 98.60 | 98.88 | 95.20 | 97.38 | 98.50 | 98.50 | 98.09 | 96.58 | 100.00 | 100.00 | 100.00 | 99.00 | 98.83 | 98.69 | 99.01 | | |
| | fake | 99.92 | 88.20 | 99.30 | 98.35 | 50.15 | 62.60 | 39.10 | 76.80 | 19.35 | 17.80 | 0.00 | 6.21 | 3.00 | 23.24 | 5.98 | 10.80 | | |
| | total | 99.75 | 93.40 | 99.09 | 96.77 | 73.80 | 80.55 | 68.80 | 87.45 | 55.03 | 47.14 | 50.00 | 20.33 | 51.00 | 61.04 | 37.22 | 45.97 | 60.26 | 66.71 |
| FreqNet | real | 100.00 | 91.00 | 86.18 | 99.80 | 90.14 | 99.50 | 99.50 | 95.16 | 100.00 | 97.00 | 97.00 | 97.00 | 93.00 | 98.05 | 96.26 | 96.90 | | |
| | fake | 99.20 | 91.40 | 99.70 | 68.63 | 94.11 | 95.80 | 43.20 | 84.58 | 4.89 | 21.67 | 2.00 | 32.62 | 10.00 | 1.95 | 4.75 | 11.13 | | |
| | total | 99.60 | 91.20 | 92.94 | 84.22 | 92.12 | 97.65 | 71.35 | 89.87 | 48.83 | 48.57 | 49.50 | 42.32 | 51.50 | 50.00 | 35.58 | 46.61 | 61.39 | 68.24 |
| ForgeLens | real | 99.65 | 95.40 | 96.96 | 83.63 | 94.41 | 90.30 | 90.30 | 92.95 | 81.53 | 84.54 | 84.04 | 84.69 | 100.00 | 98.05 | 96.07 | 89.85 | | |
| | fake | 100.00 | 99.80 | 99.98 | 100.00 | 87.45 | 98.60 | 50.80 | 90.95 | 44.24 | 67.43 | 20.41 | 8.60 | 14.74 | 42.19 | 55.39 | 36.14 | | |
| | total | 99.83 | 97.60 | 98.47 | 91.82 | 90.93 | 94.45 | 70.55 | 91.95 | 61.50 | 73.53 | 51.56 | 19.97 | 57.81 | 70.12 | 69.13 | 57.66 | 70.88 | 74.81 |
| FatFormer | real | 100.00 | 98.10 | 97.22 | 98.95 | 99.19 | 98.80 | 98.80 | 98.72 | 98.74 | 100.00 | 100.00 | 100.00 | 94.00 | 98.44 | 99.25 | 98.63 | | |
| | fake | 99.78 | 99.85 | 100.00 | 100.00 | 84.88 | 98.10 | 56.30 | 91.27 | 3.59 | 48.33 | 0.00 | 13.48 | 12.00 | 43.16 | 21.75 | 20.33 | | |
| | total | 99.89 | 98.98 | 98.61 | 99.47 | 92.04 | 98.45 | 77.55 | 95.00 | 47.55 | 66.79 | 50.00 | 26.51 | 53.00 | 70.80 | 47.86 | 51.79 | 67.03 | 73.39 |
| Trufor | real | 93.98 | 91.85 | 93.02 | 83.79 | 80.94 | 81.00 | 81.00 | 86.51 | 96.12 | 95.00 | 95.00 | 95.00 | 94.00 | 93.36 | 90.84 | 94.19 | | |
| | fake | 7.78 | 15.30 | 7.34 | 54.72 | 29.20 | 13.20 | 8.80 | 19.48 | 67.93 | 98.88 | 43.00 | 13.48 | 95.00 | 35.36 | 72.84 | 60.93 | | |
| | total | 50.88 | 53.58 | 50.18 | 69.26 | 55.12 | 47.10 | 44.90 | 53.00 | 81.04 | 97.50 | 69.00 | 25.76 | 94.50 | 64.36 | 78.90 | 73.01 | 61.42 | 63.01 |
| AdaIFL | real | 85.78 | 81.00 | 89.12 | 29.36 | 95.46 | 84.70 | 84.70 | 78.59 | 96.62 | 85.00 | 85.00 | 85.00 | 99.01 | 83.79 | 92.52 | 89.56 | | |
| | fake | 15.30 | 15.30 | 10.30 | 26.46 | 12.79 | 22.40 | 17.80 | 17.19 | 81.74 | 86.11 | 72.00 | 57.62 | 47.00 | 43.36 | 50.59 | 59.77 | | |
| | total | 50.54 | 48.15 | 49.71 | 27.89 | 54.19 | 53.55 | 51.25 | 47.90 | 88.66 | 85.71 | 78.50 | 61.75 | 73.13 | 63.57 | 51.45 | 71.82 | 57.47 | 59.86 |
| HiFi-Net | real | 74.75 | 97.00 | 97.52 | 97.70 | 77.80 | 92.20 | 92.20 | 89.88 | 97.25 | 96.00 | 96.00 | 96.00 | 100.00 | 98.83 | 98.88 | 97.57 | | |
| | fake | 98.95 | 82.50 | 40.48 | 98.35 | 39.77 | 75.90 | 53.30 | 69.89 | 2.17 | 13.89 | 22.00 | 23.05 | 1.00 | 20.70 | 41.79 | 17.80 | | |
| | total | 86.85 | 89.75 | 69.00 | 98.02 | 58.82 | 84.05 | 72.75 | 79.89 | 46.40 | 43.21 | 59.00 | 34.04 | 50.50 | 59.77 | 61.02 | 50.56 | 61.93 | 65.23 |
| Trufor-FT | real | 86.98 | 84.10 | 85.68 | 48.17 | 78.06 | 79.10 | 79.10 | 77.31 | 77.35 | 92.00 | 92.00 | 92.00 | 74.00 | 86.52 | 85.23 | 85.59 | | |
| | fake | 12.42 | 22.50 | 15.60 | 59.33 | 15.75 | 17.70 | 17.50 | 22.97 | 19.78 | 30.00 | 11.00 | 28.01 | 20.00 | 27.93 | 15.67 | 21.77 | | |
| | total | 49.70 | 53.30 | 50.64 | 53.75 | 46.96 | 48.40 | 48.30 | 50.15 | 51.46 | 52.14 | 52.00 | 37.65 | 47.00 | 57.23 | 39.11 | 48.01 | 49.06 | 49.08 |
| AdaIFL-FT | real | 95.60 | 93.50 | 94.40 | 97.55 | 94.50 | 87.00 | 87.00 | 92.79 | 97.25 | 90.00 | 90.00 | 90.00 | 86.14 | 91.02 | 93.64 | 91.15 | | |
| | fake | 3.05 | 4.85 | 3.52 | 1.65 | 5.63 | 6.60 | 2.90 | 4.03 | 33.37 | 84.44 | 28.00 | 46.99 | 52.00 | 26.76 | 23.65 | 42.17 | | |
| | total | 49.33 | 49.18 | 48.96 | 49.60 | 50.14 | 46.80 | 44.95 | 48.42 | 63.08 | 86.43 | 59.00 | 53.46 | 69.15 | 58.89 | 47.23 | 62.46 | 54.55 | 55.44 |
| UniFD-FT | real | 98.40 | 95.80 | 97.02 | 87.09 | 91.98 | 96.40 | 96.40 | 94.73 | 93.67 | 100.00 | 100.00 | 100.00 | 99.00 | 97.46 | 97.38 | 98.22 | | |
| | fake | 99.95 | 93.70 | 99.88 | 99.30 | 65.27 | 75.20 | 52.00 | 83.61 | 31.85 | 32.78 | 0.00 | 10.64 | 5.00 | 35.35 | 11.40 | 18.15 | | |
| | total | 99.18 | 94.75 | 98.45 | 93.20 | 78.65 | 85.80 | 74.20 | 89.18 | 60.41 | 56.76 | 50.00 | 24.10 | 52.00 | 66.41 | 40.37 | 50.01 | 64.08 | 69.59 |
| FreqNet-FT | real | 100.00 | 90.95 | 86.18 | 99.80 | 90.14 | 99.50 | 99.50 | 95.15 | 96.84 | 94.00 | 94.00 | 94.00 | 93.00 | 94.53 | 99.25 | 95.09 | | |
| | fake | 99.20 | 91.40 | 99.70 | 68.73 | 94.18 | 95.80 | 43.20 | 84.60 | 10.11 | 36.11 | 8.00 | 33.51 | 12.00 | 7.62 | 12.92 | 17.18 | | |
| | total | 99.60 | 91.17 | 92.94 | 84.27 | 92.16 | 97.65 | 71.35 | 89.88 | 50.18 | 56.79 | 51.00 | 42.62 | 52.00 | 51.07 | 56.12 | 50.50 | 63.74 | 69.63 |
| ForgeLens-FT | real | 99.50 | 77.55 | 92.00 | 38.52 | 48.32 | 77.50 | 77.50 | 72.98 | 93.16 | 93.00 | 93.00 | 93.00 | 42.57 | 93.95 | 87.85 | 85.22 | | |
| | fake | 99.98 | 100.00 | 99.98 | 100.00 | 95.81 | 99.90 | 57.80 | 93.35 | 48.80 | 86.67 | 13.00 | 16.31 | 83.00 | 48.83 | 56.89 | 50.50 | | |
| | total | 99.74 | 88.78 | 95.99 | 69.26 | 72.03 | 88.70 | 67.65 | 83.16 | 69.30 | 88.93 | 53.00 | 27.86 | 62.69 | 71.39 | 67.32 | 62.93 | 71.64 | 73.05 |
| FatFormer-FT | real | 99.58 | 88.10 | 83.10 | 56.73 | 13.48 | 84.50 | 84.50 | 72.86 | 94.88 | 94.00 | 94.00 | 94.00 | 34.00 | 91.21 | 92.34 | 84.92 | | |
| | fake | 99.95 | 99.95 | 100.00 | 100.00 | 99.44 | 99.30 | 73.70 | 96.05 | 50.54 | 91.67 | 12.00 | 47.52 | 84.00 | 63.67 | 45.01 | 56.34 | | |
| | total | 99.76 | 94.03 | 91.55 | 78.36 | 56.39 | 91.90 | 79.10 | 84.44 | 71.16 | 92.50 | 53.00 | 54.52 | 59.00 | 77.44 | 60.66 | 66.94 | 74.68 | 75.69 |
| DeFake (Ours) | real | 97.92 | 83.55 | 75.08 | 63.08 | 34.06 | 83.80 | 83.80 | 74.47 | 86.62 | 92.00 | 92.00 | 92.00 | 60.00 | 87.50 | 85.61 | 85.10 | | |
| | fake | 99.80 | 99.90 | 100.00 | 100.00 | 98.15 | 99.40 | 82.80 | 97.15 | 55.33 | 93.89 | 14.00 | 54.26 | 87.00 | 67.19 | 60.02 | 61.67 | | |
| | total | 98.96 | 91.72 | 87.54 | 81.54 | 66.05 | 91.60 | 83.30 | 85.82 | 69.88 | 93.21 | 53.00 | 59.94 | 73.50 | 77.34 | 68.64 | 70.79 | **77.58** | **78.30** |

Table 6: Detailed comparison results with state-of-the-art methods across both the tasks. Here, *real*, *fake* and *total* denotes the real, fake and overall accuracies for all the datasets.

| Methods | Acc (%) | **Fully Synthetic image datasets** | | | | | | | | **Partially Manipulated image datasets** | | | | | | | | HM↑ | Avg (T)↑ |
|---|---|---|---|---|---|---|---|---|---|---|---|---|---|---|---|---|---|---|---|
| | | ProGAN | BigGAN | GauGAN | StarGAN | Deepfake | DALLE | Guided | Avg (F) | CASIA1 | Columbia | Coverage | NIST16 | DSO-1 | CocoGLIDE | MagicBrush | Avg (P) | | |
| UniFD | real | 99.58 | 98.60 | 98.88 | 95.20 | 97.38 | 98.50 | 98.50 | 98.09 | 96.58 | 100.00 | 100.00 | 100.00 | 99.00 | 98.83 | 98.69 | 99.01 | | |
| | fake | 99.92 | 88.20 | 99.30 | 98.35 | 50.15 | 62.60 | 39.10 | 76.80 | 19.35 | 17.80 | 0.00 | 6.21 | 3.00 | 23.24 | 5.98 | 10.80 | | |
| | total | 99.75 | 93.40 | 99.09 | 96.77 | 73.80 | 80.55 | 68.80 | 87.45 | 55.03 | 47.14 | 50.00 | 20.33 | 51.00 | 61.04 | 37.22 | 45.97 | 60.26 | 66.71 |
| UniFD-FT | real | 98.40 | 95.80 | 97.02 | 87.09 | 91.98 | 96.40 | 96.40 | 94.73 | 93.67 | 100.00 | 100.00 | 100.00 | 99.00 | 97.46 | 97.38 | 98.22 | | |
| | fake | 99.95 | 93.70 | 99.88 | 99.30 | 65.27 | 75.20 | 52.00 | 83.61 | 31.85 | 32.78 | 0.00 | 10.64 | 5.00 | 35.35 | 11.40 | 18.15 | | |
| | total | 99.18 | 94.75 | 98.45 | 93.20 | 78.65 | 85.80 | 74.20 | 89.18 | 60.41 | 56.76 | 50.00 | 24.10 | 52.00 | 66.41 | 40.37 | 50.01 | 64.08 | 69.59 |
| UniFD + DeFake | real | 96.02 | 90.95 | 93.28 | 75.09 | 78.28 | 89.50 | 89.50 | 87.52 | 90.76 | 98.00 | 98.00 | 98.00 | 97.03 | 91.80 | 94.77 | 95.48 | | |
| | fake | 100.00 | 96.10 | 99.94 | 99.65 | 80.54 | 83.60 | 63.10 | 88.99 | 44.78 | 45.00 | 4.00 | 12.41 | 18.00 | 42.97 | 18.42 | 26.51 | | |
| | total | 98.01 | 93.52 | 96.61 | 87.37 | 79.41 | 87.53 | 80.70 | 89.02 | 66.02 | 63.93 | 51.00 | 25.30 | 57.71 | 67.38 | 44.14 | 53.64 | **66.94** | **71.33** |
| ForgeLens | real | 99.65 | 95.40 | 96.96 | 83.63 | 94.41 | 90.30 | 90.30 | 92.95 | 81.53 | 84.54 | 84.04 | 84.69 | 100.00 | 98.05 | 96.07 | 89.85 | | |
| | fake | 100.00 | 99.80 | 99.98 | 100.00 | 87.45 | 98.60 | 50.80 | 90.95 | 44.24 | 67.43 | 20.41 | 8.60 | 14.74 | 42.19 | 55.39 | 36.14 | | |
| | total | 99.83 | 97.60 | 98.47 | 91.82 | 90.93 | 94.45 | 70.55 | 91.95 | 61.50 | 73.53 | 51.56 | 19.97 | 57.81 | 70.12 | 69.13 | 57.66 | 70.88 | 74.81 |
| ForgeLens-FT | real | 99.50 | 77.55 | 92.00 | 38.52 | 48.32 | 77.50 | 77.50 | 72.98 | 93.16 | 93.00 | 93.00 | 93.00 | 42.57 | 93.95 | 87.85 | 85.22 | | |
| | fake | 99.98 | 100.00 | 99.98 | 100.00 | 95.81 | 99.90 | 57.80 | 93.35 | 48.80 | 86.67 | 13.00 | 16.31 | 83.00 | 48.83 | 56.89 | 50.50 | | |
| | total | 99.74 | 88.78 | 95.99 | 69.26 | 72.03 | 88.70 | 67.65 | 83.16 | 69.30 | 88.93 | 53.00 | 27.86 | 62.69 | 71.39 | 67.32 | 62.93 | 71.64 | 73.05 |
| ForgeLens + DeFake | real | 99.35 | 93.20 | 94.06 | 72.14 | 83.56 | 83.50 | 83.70 | 87.07 | 84.56 | 80.00 | 81.00 | 80.00 | 82.18 | 97.66 | 94.77 | 85.94 | | |
| | fake | 100.00 | 99.90 | 99.98 | 100.00 | 91.40 | 99.00 | 56.50 | 92.40 | 52.39 | 88.89 | 30.00 | 15.43 | 69.00 | 50.00 | 65.62 | 53.16 | | |
| | total | 99.67 | 96.55 | 97.02 | 86.07 | 87.47 | 91.25 | 70.10 | 89.73 | 67.25 | 85.71 | 55.50 | 25.15 | 75.62 | 73.83 | 75.44 | 65.21 | **75.53** | **77.47** |

Table 7: Generalization to other CLIP-based base models. The detailed comparison results with real, fake and total accuracies are reported. FT denotes standard fine-tuning, while *+DeFake* denotes adaptation using our proposed modules. Addition of the proposed modules outperforms both the zero-shot and FT performances as shown in bold.

