# OpenReview forum: "DeFake: Data-Efficient Adaptation for Generalized Deepfake Detection"
_ICLR.cc/2026/Conference — ICLR 2026 Conference Withdrawn Submission_

### Official Review · Reviewer_rMGU · 2025-10-20

**Soundness:** 2
**Presentation:** 2
**Contribution:** 1
**Rating:** 2
**Confidence:** 5

**Summary:**

The paper proposes DeFake (Data-Efficient Adaptation for Generalized Deepfake Detection), a unified framework capable of detecting both fully synthetic and partially manipulated images using limited training data. By adapting a pretrained synthetic image detector, DeFake maintains performance across both tasks without requiring large datasets. It introduces three key modules—Noise-Aware Patch Enhancement (NPE) for local artifact amplification, Adaptive Score Aggregation (ASA) for balancing global and local cues, and Multi-Scale Alignment (MSA) for improved feature discrimination. Experiments on 14 datasets show that DeFake outperforms existing methods in both detection settings.

**Strengths:**

1. Traditional partial image manipulation detection models are typically pretrained on large-scale computer vision datasets such as ImageNet. In contrast, the authors pretrain DeFake using ProGAN-generated deepfake images, which appears to be a more intuitive and task-aligned approach for forgery detection, as it exposes the model to synthetic generation artifacts rather than generic object semantics.
2. The claim that DeFake, fine-tuned on only 128 images (64 real and 64 fake) on top of pretrained ProGAN-generated synthetic images, outperforms state-of-the-art partial image manipulation models is quite ambitious. This result warrants stronger empirical justification through additional experiments and analysis to establish its credibility.
3. The qualitative analysis using t-SNE visualizations for partial manipulation detection is commendable and provides valuable insights. This approach should be encouraged and adopted by other researchers, particularly when presented alongside binary mask localizations for clearer interpretability.

**Weaknesses:**

Major:
1. The probabilistic formulation in Eq. (1), which models partially manipulated images as a mixture of fully synthetic and real image distributions parameterized by λ, appears conceptually flawed. Fully synthetic images (Df) and partially manipulated images (Dp) arise from fundamentally different generative processes—global generation versus localized deterministic edits. A convex mixture of image-level densities with a scalar λ overlooks the deterministic compositing operation and the known binary mask used to create Dp. In practice, a partially manipulated image is produced by X = Φ(R, F, M), where R ∼ Dr, F is a locally generated, inserted, or edited region (e.g., diffusion inpainting or splicing), M is the binary mask, and Φ denotes the compositing function. Under this process, P(X | Dp) is governed by the joint distribution p(R) p(F | R, edit specification) p(M) and the operator Φ, rather than by a simple convex combination of P(· | Df) and P(· | Dr). Moreover, since the manipulated regions and corresponding masks are explicitly known during generation and training, representing this as a probabilistic mixture oversimplifies the underlying mechanism and weakens the theoretical grounding of the proposed motivation. How can the authors justify representing the distribution of partially manipulated images as a convex combination of fully synthetic and real image distributions through a single scalar parameter λ, given that these arise from fundamentally different processes? Does this formulation overlook the role of localized compositing or blending operations inherent in partial image manipulations?
2. The authors list the Noise-Aware Patch Enhancement (NPE) module as a key contribution. However, it appears to primarily combine Noiseprint++ features from TruFor with a FatFormer-based CLIP encoder. Could the authors clarify which component or design choice within NPE represents the actual novel contribution beyond integrating existing feature extractors? Similarly, the claimed novelty of the Adaptive Score Aggregation (ASA) module is unclear, as the CLIP similarity score mechanism appears directly adapted from FatFormer. Moreover, the use of patch-wise contrastive loss in the MSA module is not new to the image manipulation community, with prior works such as NCL-IML (ICCV 2023), CFL-Net (WACV 2023), and Forensics Adapter (CVPR 2025) employing similar objectives. Overall, the reviewer finds the architectural novelty of the proposed model insufficiently justified.
3. The authors are requested to clarify the reported evaluation metrics—Accuracy and Harmonic Mean. What exactly does “Accuracy” represent in this context? Is it computed at the image level or pixel level, and does it correspond to a real vs. fake classification score for fully synthetic image detection? How is this accuracy metric computed and compared for partially manipulated image detection? Additionally, the term “Harmonic Mean” requires clarification—which metrics is this harmonic mean calculated (e.g., Accuracy, AUC, F1, or IoU)? The authors are also encouraged to report standard metrics such as F1, AUC, and IoU (both image-level and pixel-level), which are widely adopted in the image manipulation localization literature, to enable a fair comparison of DeFake with existing baselines.
4. The authors state in Appendix A.1 (Lines 619–647) that only 128 images (64 real and 64 fake) were used for fine-tuning DeFake, with pretraining performed on ProGAN-generated synthetic images across four classes (car, cat, chair, horse). Could the authors clarify how many synthetic images were used for pretraining, and whether a different CLIP-based fully synthetic detection model was employed during this stage? It is unclear how pretraining on only four object classes enables strong generalization to unseen and diverse categories present in the partial manipulation datasets (CASIA, Columbia, NIST16, Coverage, COCOGlide, MagicBrush). Is this performance attributable to specific low-level cues captured by the backbone network—such as frequency, JPEG compression, or noise features? The paper lacks sufficient details regarding both training and pretraining setups, which are critical for reproducibility and for understanding the source of DeFake’s reported generalization.
5. The Limitations and Future Scope section (Lines 476–479) is too brief and lacks depth. The authors only mention the absence of explicit spatial localization and the limited performance in large-scale training scenarios. However, there is no discussion of model robustness, failure cases, or future directions to address them. The paper would benefit from including analyses of robustness to common perturbations (e.g., compression, blur, resizing, illumination changes, unseen manipulation types) and examples where DeFake fails to generalize. A more comprehensive limitations section outlining these aspects and concrete plans for improvement would strengthen the paper’s credibility and transparency.
6. In Figure 4, the fine-tuned model shows lower accuracy but a higher harmonic mean compared to the zero-shot setting for ForgeLens. However, the authors do not provide any explanation or analysis for this discrepancy. Lines 467–470 merely state that DeFake performs better than both zero-shot and fine-tuned settings without any supporting discussion. The authors are encouraged to include an analysis explaining why fine-tuning leads to this trade-off between accuracy and harmonic mean, and what it implies about the model’s generalization or calibration behavior.

Minor:
1. Lines 95–107 and the contribution section (Lines 108–116) convey overlapping information about the paper. Consider merging them to avoid redundancy and use the freed space to report additional experimental results or analysis.
2. Please include the performance of DeFake alongside the SOTA methods in Figure 1 to substantiate the t-SNE claim—that is, the observed overlap between partially manipulated and real images indicating limited generalization—and to demonstrate how DeFake improves this generalization. Additionally, consider adding mask visualizations for the partially manipulated samples to improve interpretability and reader clarity.
3. The font of Table 1 is very small. Can it be increased for better readability? Mention the metric reported in the caption.
4. Could the authors include the binary predicted masks from DeFake in Figure 5 to compare them against the ground truth masks visually? Additionally, please provide t-SNE visualizations of patch-level representations for other baseline SOTA methods on the same set of images. This would offer stronger evidence of the claimed improvements achieved by DeFake in terms of feature separability and localization quality.

**Questions:**

1. Did the authors consider evaluation on fully AI-generated synthetic content datasets? Recent studies [2,3,4] have shown that traditional artifacts associated with splicing, copy-move, and removal forgeries are significantly diminished in AI-generated images, making them difficult to detect using conventional partial manipulation models. It is unclear whether the proposed method accounts for this distributional shift. Furthermore, while the authors report results on COCOGlide and MagicBrush, a fairer comparison would include more recent benchmarks such as [1], which also address the detection of both partially manipulated and fully synthetic images for deepfake detection. The authors are encouraged to compare their approach against [1] rather than relying solely on HiFi-Net (CVPR 2023), which represents an earlier generation of methods in this domain.
2. The authors claim that the proposed method addresses generalized deepfake detection, yet no evaluation has been conducted on standard deepfake detection benchmarks such as FaceForensics++ (FF++), CelebDF, DeepForensics, or HiFiFace. To substantiate the generalization claims, the authors are strongly encouraged to include experiments on these widely used datasets or provide a clear justification for their exclusion.
3. The authors utilize Noiseprint++ features extracted from input images. To better establish the contribution of the Noise-Aware Patch Enhancement (NPE) module, the authors are encouraged to conduct ablation experiments using alternative feature representations—such as plain RGB features, frequency-domain features, SRM filters, Bayar convolution, and DCT-based features—and compare their impact on performance. This analysis would clarify whether the observed improvements stem from the NPE design itself or from the choice of the underlying noise-based features.


[1] Kundu et al., Towards a universal synthetic video detector: From face or background manipulations to fully AI-generated content, CVPR 2025

[2] Sida: Social media image deepfake detection, localization and explanation with large multimodal model, CVPR 2025

[3] OpenSDI: Spotting Diffusion-Generated Images in the Open World, CVPR 2025

[4] Rethinking Image Editing Detection in the Era of Generative AI Revolution, ACM MM 2024

**Details Of Ethics Concerns:**

The paper lacks any discussion on ethical considerations or reproducibility, which is particularly important in the context of deepfake and image forgery detection research. Models trained for deepfake detection inherently involve sensitive domains such as human identity, biometric data, and synthetic content generation, raising potential concerns regarding bias, misuse, and false attribution. Without transparency on dataset composition, demographic balance, or bias mitigation, there is a risk that the model may exhibit unfair performance across different groups or contexts. Additionally, the paper does not discuss privacy and security implications—for instance, how the proposed detection system might handle or store manipulated images containing identifiable individuals, or how it could be safeguarded against adversarial misuse (e.g., evasion or re-identification attacks). A clear statement addressing ethical safeguards, data handling practices, and responsible disclosure would strengthen the paper’s alignment with ethical standards. Finally, reproducibility details such as preprocessing steps, and implementation settings are insufficiently described, limiting verification and independent validation of the results. Given the societal impact and security relevance of deepfake research, the absence of these discussions warrants an ethics review flag.

---

### Official Review · Reviewer_viox · 2025-10-30

**Soundness:** 3
**Presentation:** 3
**Contribution:** 3
**Rating:** 4
**Confidence:** 4

**Summary:**

This paper presents a framework to unify the detection of two classes of manipulated media: fully synthetic images and partially manipulated images.

**Strengths:**

The motivation to tackle these two tasks under a single umbrella is timely and addresses a significant need within the forensics community. The paper is well-structured, and the presentation is clear.

**Weaknesses:**

However, I have several major concerns regarding the novelty of the proposed method, the practical significance of the claimed contributions, and the trade-offs observed in the experimental results. My detailed comments are listed below:

1. The authors emphasize that their method is "data-efficient," particularly for the partially manipulated data. From my perspective, the practical significance of this contribution is limited. Partially manipulated data is not exceptionally difficult to acquire; one can generate a large dataset through manual editing or, more scalably, through on-the-fly augmentation during the training process. A more compelling argument for this contribution would be to show that using the proposed approach with limited partial data significantly outperforms baselines that have access to a larger, easily generated dataset. Furthermore, if incorporating more partially manipulated data (even if generated) could further boost the model's performance, it would strengthen the paper's claims. Since the authors do not fine-tune the CLIP backbone, the training efficiency is already relatively high, which slightly diminishes the perceived impact of this data-efficiency aspect from a technical standpoint.

2. The technical novelty of the proposed framework appears to be an integration of several existing techniques. The core components, such as (a) adapter-based fine-tuning for large vision models like CLIP, (b) the fusion of RGB and noise-based views for forgery detection, and (c) the use of local-global contrastive learning, have been thoroughly investigated in prior work. While the combination of these elements for this specific unified task is new, the paper reads more as a thoughtful engineering effort than a proposal of an originally novel technique.

3. The results in Table 2 are concerning. The proposed unified model, Full DeFake, appears to achieve improved performance on partially manipulated images at the cost of degraded performance on fully synthetic ones. This trade-off raises a critical question: in a practical application, why wouldn't one simply deploy two separate, specialized models, each optimized for its respective task, to achieve the best performance on both? A truly effective unified framework should ideally demonstrate synergistic benefits (i.e., mutual promotion), where the joint training leads to improvements on both tasks, or at the very least, maintains state-of-the-art performance on all sub-tasks without compromise. The current results do not provide evidence of such synergy and instead suggest a performance compromise, which weakens the argument for a unified model.

4. The paper does not include an evaluation of the model's robustness against common post-processing operations, such as compression, resizing, or noise addition. These operations are prevalent in real-world scenarios where images are shared across social media platforms. The performance of a forgery detector under such conditions is critical for practical utility.

5. In Figure 3, the "Language-guidance block" is depicted, but its inputs are not clearly illustrated. It is unclear how language prompts or other forms of guidance are fed into this block. Please revise the figure to provide a complete and more intuitive illustration of the data flow.

**Questions:**

See Weaknesses.

---

### Official Review · Reviewer_ow7y · 2025-10-31

**Soundness:** 2
**Presentation:** 2
**Contribution:** 2
**Rating:** 2
**Confidence:** 3

**Summary:**

This paper proposes a new generalized deepfake detection method “DeFake” that looks at tackling the problem of not only determining fully synthetic images but also partially manipulated images. Their method utilizes a pretrained fully synthetic image detector (e.g. FatFormer which is a CLIP based base model) which can be extended to other methods like Uniformer and ForgeLens which they showcase in their paper. Their method introduces three lightweight, generalizable modules namely Noise-Patch Enhancement (NPE), Adaptive Score Aggregation (ASA) and Multi-Scale Alignment (MSA).

Using this method they showcase how their method finetuned on a small amount of training images can outperform current methods when looking at both tasks (fully synthetic and partially manipulated images) it achieves the highest average accuracy of 78.30% and harmonic mean (HM) of 77.58%. This is done across 14 different datasets ranging from fully synthetic image datasets like ProGAN, BigGAN and partially manipulated image datasets like CAISA, MagicBrush and CocoGLIDE.

**Strengths:**

* This paper tackles the challenging problem of not only detecting synthetic images but also partially manipulated images at the same time

* The paper showcases how CLIP trained models can be used to align to partially manipulated image domain

**Weaknesses:**

* A major aspect of this evaluation seems to be missing. Namely, the way this training is set up it seems to focus on training on a small subset of images that helps showcase a high performance for the DeFake model. It raises the question why the remaining training images were disregarded since most of the other models typically require a larger amount of training images to adjust its parameters to the new domain [1,2]. There is no indication of what the upper bound is for this training paradigm as most of the training images were disregarded, hence we do not know if fine tuning the other models with the left over training images would affect performance. Additionally what occurs if you train using different levels of training data, e.g. 50%, 100%.

[1] Trufor: Leveraging all-round clues for trustworthy image forgery detection and localization, CVPR 2023

[2] Mvss-net: Multi-view multiscale supervised networks for image manipulation detection (IEEE 2022)

* Utilizes a small amount of partial manipulation detection models when other works are present [1,2,3,4,5]

[1] FakeShield: Explainable Image Forgery Detection and Localization via Multi-modal Large Language Models (ICLR 2025)

[2] ObjectFormer for Image Manipulation Detection and Localization (CVPR 2022)

[3] PSCC-Net: Progressive spatio-channel correlation network for image manipulation detection and localization (IEEE 2022)

[4] OpenSDI: Spotting Diffusion-Generated Images in the Open World [CVPR 2025)

[5] Mvss-net: Multi-view multiscale supervised networks for image manipulation detection (IEEE 2022)

* More recent clip based models in the partial manipulation detection model is ignored [1]

[1] OpenSDI: Spotting Diffusion-Generated Images in the Open World [CVPR 2025)

* Additionally the paper does not look at larger datasets that utilize diffusion based inpaintings like OpenSDI, GRE, SIDA and DOLOS to evaluate on [1,2,3,4]

[1] OpenSDI: Spotting Diffusion-Generated Images in the Open World [CVPR 2025)

[2] Rethinking Image Editing Detection in the Era of Generative AI Revolution (ACM MM, 2024)

[3] Sida: Social media image deepfake detection, localization and explanation with large multimodal model. (CVPR 2025)

[4] Weakly-supervised deepfake localization in diffusion-generated images (WACV 2024)

* The paper relies on the backbone of established fully synthetic detectors like FatFormer and ForgeLens which utilize a CLIP backbone and NoisePrint model which makes the contribution not appear as novel.

**Questions:**

* Not much exploration is done in terms of visualization of features extracted, for instance using CAM. What did the features look like before implementing the method, this was showcased in ForgeLens but not shown for DeFake.

---

### Official Review · Reviewer_toV4 · 2025-11-01

**Soundness:** 4
**Presentation:** 3
**Contribution:** 3
**Rating:** 4
**Confidence:** 4

**Summary:**

This paper presents a generalized synthesized or manipulated (deepfake) detection method, Data-Efficient Adaptation for Generalized Deepfake Detection (DeFake)) that can cover both fully synthetic and partially manipulated images aiming at more practical application. One of the advantages of the proposed method is that it requires only limited training samples (128) while achieving better performance than previous works.

The key contributions in this paper are the following three modules: (a) Noise-aware Patch Enhancement (NPE) that captures manipulation artifacts present in partially manipulated images, (b) Adaptive Score Aggregation (ASA) that dynamically updates the importance of the global image-level semantics and the local patch-level artifacts, and (c) Contrastive learning for multi-scale alignment at both image and patch-level.

Experiments using a large number of datasets demonstrate the effectiveness and validity of the proposed DeFake.

**Strengths:**

As shown in Figure 2, the ability to detect both fully synthesized and partially manipulated is difficult, and the proposed method is balanced for both types of deepfakes.

**Weaknesses:**

Figures and texts in those figures, and texts in tables are too small. I understand that the paper can be enlarged, but visibility of the figures and tables are critically poor in the current manuscript.

This paper claims that 128 images are good enough for training, and it is shown that 128 images are necessary and sufficient in Table 4. But how can we generalize this for future extensions? Can the authors explain why 128 images are good enough? Does the optimal value change depending on the other parameters or model architecture?

How to determine the hyperparameters such as $tau and $\lambda_1 - $\lambda_3 is not clear. I conjecture that these parameters are decided by empirical study. The authors may want to show the sensitivity of these parameters by changing the values (I can see some cases in Table 3, but I am requesting a graph to show the sensitivity, not a table).

Robustness to disturbances such as noise addition, image filtering, JPEG compression, etc. is not discussed. It is clear that attackers would to something to delete the clue of the synthesis or manipulation as much as possible before distributing it. Therefore, such discussion on robustness is important.

More detailed case studies, especially failure cases, would help possible readers to understand the proposed method better.

Some minor modification proposals (no need to reply)
- missing commas and periods after equations
- no comma is needed after “where” when explaining the equation.
- “cos” in equations should be “\cos”
- The are some grammatical mistakes such as “with slight increase” even though the overall writing quality is good.

**Questions:**

None

---

> ### Comment · Reviewer_toV4 · 2025-11-27
> **no rebuttal comments so far**
>
> No rebuttal comments are provided by the authors so far.
> So, no reason for me to change the score.

---

### Note · Authors · 2025-12-08

I have read and agree with the venue's withdrawal policy on behalf of myself and my co-authors.